# The Effect of Post Deposition Treatment on Properties of ALD Al-Doped ZnO Films

**DOI:** 10.3390/nano13050800

**Published:** 2023-02-22

**Authors:** Dimitrina Petrova, Blagovest Napoleonov, Chau Nguyen Hong Minh, Vera Marinova, Yu-Pin Lan, Ivalina Avramova, Stefan Petrov, Blagoy Blagoev, Vladimira Videva, Velichka Strijkova, Ivan Kostadinov, Shiuan-Huei Lin, Dimitre Dimitrov

**Affiliations:** 1Faculty of Engineering, South-West University “Neofit Rilski”, 2700 Blagoevgrad, Bulgaria; 2Institute of Optical Materials and Technologies, Bulgarian Academy of Sciences, 1113 Sofia, Bulgaria; 3Department of Electrophysics, National Yang Ming Chiao Tung University, Hsinchu 30010, Taiwan; 4College of Photonics, National Yang Ming Chiao Tung University, Tainan 75110, Taiwan; 5Institute of General and Inorganic Chemistry, Bulgarian Academy of Sciences, 1113 Sofia, Bulgaria; 6Institute of Solid State Physics, Bulgarian Academy of Sciences, 1784 Sofia, Bulgaria; 7Faculty of Chemistry and Pharmacy, Sofia University, 1 James Bourchier Blvd., 1164 Sofia, Bulgaria

**Keywords:** aluminum-doped ZnO, ALD, transparent conducting oxide, XPS, oxygen vacancies, UV–ozone

## Abstract

In this paper, aluminum-doped zinc oxide (ZnO:Al or AZO) thin films are grown using atomic layer deposition (ALD) and the influence of postdeposition UV–ozone and thermal annealing treatments on the films’ properties are investigated. X-ray diffraction (XRD) revealed a polycrystalline wurtzite structure with a preferable (100) orientation. The crystal size increase after the thermal annealing is observed while UV–ozone exposure led to no significant change in crystallinity. The results of the X-ray photoelectron spectroscopy (XPS) analyses show that a higher amount of oxygen vacancies exists in the ZnO:Al after UV–ozone treatment, and that the ZnO:Al, after annealing, has a lower amount of oxygen vacancies. Important and practical applications of ZnO:Al (such as transparent conductive oxide layer) were found, and its electrical and optical properties demonstrate high tunability after postdeposition treatment, particularly after UV–Ozone exposure, offers a noninvasive and easy way to lower the sheet resistance values. At the same time, UV–Ozone treatment did not cause any significant changes to the polycrystalline structure, surface morphology, or optical properties of the AZO films.

## 1. Introduction

Transparent conducting oxides (TCO) are used as functional layers in various devices, such as solar cells, flat panel displays, organic light-emitting diodes (OLED), liquid crystal (LC) devices, etc. [1,2,3]. Moreover, TCOs prepared using specific fabrication processes with low optical losses, tunable optical properties, and metal-like conductivity have found further applications in complementary metal–oxide–semiconductors (CMOSs), plasmonics, metamaterials, and nonlinear optics [4]. Currently, indium tin oxide (ITO) is the most used TCO, which yields a low resistivity of 10^−4^ Ω.cm, has a transmittance higher than 85%, and possesses very good etchability [5]. However, the scarce resources and toxic nature of indium and fragility of ITO have motivated researchers to develop alternative TCO materials in order to replace ITO [6,7].

Al-doped ZnO (AZO) is one of the most viable alternative candidates for the next-generation TCO since both Al (Al_2_O_3_) and ZnO are abundant and nontoxic materials [7]. Moreover, AZO films can be easily etched to form microelectrodes for optoelectronic and display devices, possess high transparency and conductivity, and are characterized with excellent thermal stability and mechanical flexibility [8]. Numerous technologies such as spray pyrolysis [7], atomic layer deposition (ALD) [9], magnetron sputtering [10], chemical vapor deposition [11], and pulsed laser deposition [12] have been adopted to deposit AZO films. The films’ properties are sensitive to the deposition methods and process parameters. ALD allows for the preparation of uniform and conformal films on broad areas at low growth temperatures. Furthermore, the thickness of the film, which is important for numerous applications, can be controlled precisely. The influence of the ALD process parameters, such as growth temperature, purge times, and the precursor exposure length of the properties of AZO, have been reported numerous times [13,14,15,16,17].

Recently, it has been found that the removal of oxygen vacancies in AZO thin films leads to reduced carrier concentration and the Fermi level shifting toward the valence band, resulting in the work function increase and a decrease in the bandgap [18,19,20,21]. Moreover, it has been stated that cleaning AZO thin films using acetone and UV–ozone sequence enhances the work function by reducing the carbon contamination or through applying stoichiometry modifications to the thin film surface [22]. However, there are seldom reports on property modification of AZO films grown using ALD after various post-treatments.

As AZO is not a stoichiometric compound, its properties are largely dependent on deposition parameters and postdeposition treatment. In this work, AZO films were deposited using an ALD technique on soda–lime–glass (SLG) substrates. The films were exposed to UV–ozone as well as annealed at 600 °C in an air atmosphere. The effects of these postdeposition treatments on the film structure, the elements’ chemical states, the optical and electrical properties, as well as on wettability, were investigated. The post-treatment results reveal the tunability of the electrical and optical properties and indicate the significant role of the modified AZO films’ properties for ITO free optoelectronics and the future of the high-resolution liquid crystal (LC) displays industry.

## 2. Materials and Methods

AZO films were prepared using an ALD technique on the Beneq TFS-200 system (Beneq Group, Espoo, Finland) onto glass at a deposition temperature of 180 °C. In the ALD process, trimethylaluminum (TMA, Al(CH_3_)_3_), diethylzinc (DEZ, Zn(C_2_H_5_)_2_), and deionized water (H_2_O) were used as precursors for Al, Zn, and oxidant, respectively. Nitrogen was used as a carrier and purge gas at an average flow of 600 sccm. The aluminum-doping concentration in ZnO was tuned by varying the number of DEZ/H_2_O and TMA/H_2_O cycles in a standard procedure for the ALD processes [23,24,25] as follows: after 24 cycles of DEZ/H_2_O, a cycle of TMA/H_2_O was applied consisting of one so-called ‘supercycle’. The pulse durations for the precursors’ reactions were the same (200 ms) for DEZ, TMA, and H_2_O, whereas the purging time after each precursor reaction was set to 2 s.

Powder X-ray diffraction (XRD) patterns were gathered within the 2Θ range from 20 to 80° with a constant step 0.02° on a Bruker D8 Advance diffractometer with a Cu Ka radiation and LynxEye detector. Phase identification was performed with the Diffracplus EVA using the ICDD-PDF2 Database. The X-ray photoelectron spectroscopy (XPS) studies were performed in a VG Escalab MKII electron spectrometer using AlKα radiation with energy of 1486.6 eV under base pressure 10^−8^ Pa and a total instrumental resolution of 1 eV. The binding energies (BE) were determined using the O1s line (typical to ZnO) as a reference with an energy level of 530.3 eV. The accuracy of the measured BE was 0.2 eV. The photoelectron lines of the constituent elements on the surface were recorded and corrected by subtracting a Shirley-type background and quantified using the peak area and Scofield’s photoionization cross-sections. The deconvolution of spectra was performed with XPSPEAK41 software.

AFM imaging of the AZO films was performed using an Atomic Force Microscope (MFP-3D, Asylum Research, Oxford Instruments, Santa Barbara, CA USA). All measurements were taken in air and at room temperature. Silicon AFM tips (AC160TS, OLYMPUS) of 300 kHz resonance frequency and 26 N/m nominal spring constant were used. Morphometrical (roughness value) characterization was accomplished using IgorPro 6.37 software.

The AZO films’ optical transmittance spectra properties were measured using a CARY 05E UV–Vis–NIR spectrophotometer. The electrical and electric transport properties were characterized using the Hall Effect measurement system (HCS 1, Linseis GmbH Selb, Germany). Wettability property was measured with a drop shape analyzer (DSA25S, KRÜSS GmbH, Hamburg, Germany).

UV–ozone treatment (30 min, RT) and annealing (600 °C for 30 min in air) of AZO films were performed using UV Ozone Cleaner (UVC-1014, NanoBioAnalytics) and an oven. Hereinafter, AZO layers as deposited, after UV–Ozone treatment, and after thermal annealing are labeled as AZO (as deposited), AZO (UV–ozone), and AZO (annealed), respectively.

## 3. Results and Discussion

### 3.1. XRD

The XRD measurements of AZO (as deposited), AZO (UV–ozone), and AZO (annealed) were performed in the Θ–2Θ geometry and shows the hexagonal ZnO structure (wurtzite, space group P63mc) of the layers. The measurements reveal dominating (100) peaks at 2Θ between 31.79 and 31.89°, as shown in Figure 1, meaning that the films were grown preferably with the a-axis parallel to the substrate surface.

This kind of behavior has also been observed by other authors researching on AZO prepared by ALD within a specific deposition regime [9,26,27]. The studied AZO thin film samples do not show any peaks related to other phases, and this suggests that Al ions are incorporated into the wurtzite-type structure. The narrowing of the dominant (100) peak as well as the peak intensity increase after annealing indicate an improved crystallinity.

The parameters of the unit cell and the average crystallite size of AZO (as deposited), AZO (UV–ozone), and AZO (annealed) are shown in Table 1. The improved crystallinity of the annealed films should result in an increase in carrier mobility due to the reduction in scattering centers for electrons. Consequently, the large-grain AZO films showed higher conductivity and improved optical properties [28]. The (100) peak 2Θ value shifts from 31.89° to 31.79° after post-deposition annealing in air, while changing only slightly to 31.87 after UV–ozone treatment. In addition to the repair of the crystalline structure, the annealing treatment allows oxygen to diffuse from the air into the AZO films to fill the oxygen vacancies. As oxygen vacancies are known to induce stress to the lattice structure [29], a reduction in the oxygen vacancies through annealing releases the stress. The opposite occurs after UV–ozone postdeposition treatment: according to the XPS data, oxygen vacancies increase. The calculated microstrain of the as-deposited films was 7.7 × 10^−4^, which decreased to 7.5 × 10^−4^ after annealing and increased to 8.3 × 10^−4^ after UV–ozone treatment, respectively.

### 3.2. XPS

In the survey, the peaks of the XPS spectrum of the AZO films correspond to the elements of Zn, Al, O, and C, respectively (see Figure 2a). XPS analysis shows the characteristic Zn 2p doublet peak corresponding to the Zn^2+^ oxidation state [30,31]. From the Zn 2p spectra, two peaks were assigned to the Zn $2p_{3/2}$ state centered at 1022.2 eV and the Zn $\;2p_{1/2\;}$ state centered at 1044.5 eV, respectively. The peak at 1022.2 eV corresponds to the Zn-O and provides information about the zinc bonded to oxygen in the ZnO matrix [27].

The binding energy difference between these peaks is equal to approximately 23 eV, which is in agreement with the literature [32]. The Zn 2p_3/2_ peaks have a comparable full width at half maximum (FWHM) of 2.23 eV and 2.25 eV, for AZO (as deposited) and AZO (UV–ozone) films respectively, suggesting that the ozone treatment does not increase inhomogeneity in the AZO films. The same peak for annealed AZO films had an FWHM of 2.18 eV.

The XPS spectra of the O1s regions present an asymmetric peak, indicating the existence of different oxygen species (see Figure 2d–f). The O1s peaks of AZO (UV–ozone), AZO (annealed, (600 °C, 30 min)), and AZO (as deposited) are deconvoluted into three sub-peaks at around 530.3 eV, 531.3 eV, and 532.3 eV, corresponding to the O^2−^ ions bonded to the Zn^2+^ in the wurtzite structure of the hexagonal array of the ZnO (O_L_), oxygen vacancies (O_V_) in the Zn-O matrix, and chemisorbed oxygen (O_S_), respectively [33]. The concentration of the oxygen vacancies in the AZO thin films is directly related to the area of the second peak at 531.3 eV [28]. Consequently, to obtain qualitative information about oxygen vacancies, the O_v_/(O_total_) ratio was calculated. It was found that the amount of oxygen vacancies increases after UV–ozone treatment. O_v_/O_tot_ ratios for as-deposited films and films after post-treatments are shown in Table 2.

From the XPS investigations, it was found that a higher amount of oxygen vacancies exists in the AZO after UV–ozone treatment, while the AZO, after annealing, has a lower amount of oxygen vacancies.

The as-deposited AZO film’s ratio of 0.32 reduces to 0.29 after annealing, suggesting that the oxygen vacancies are filled by the oxygen from the environment (air) as a result of the annealing of the AZO films. It has been proved that the oxygen vacancies affect the optoelectronic properties of the AZO films [34] and that the variation of the O_v_/O_tot_ ratio indicates the tunability of the film properties, depending on the postdeposition treatment. Nevertheless, the exact role of the oxygen vacancies in the doping of ZnO is controversial [35]. However, it is well established that lack of oxygen correlates with higher doping. Ozone is a source of nascent oxygen that consumes oxygen vacancies, which leads to reduced doping. As a result, the surface of ozone-treated AZO is essentially “undoped” [36]. The calculated surface concentrations are a major piece of evidence for this (see Table 3). The annealing leads to a decrease in the surface contamination of AZO films as well as to an improvement in the crystal structure, which is also confirmed using XRD. However, the UV–ozone treatment leads to an increase in surface contamination, which was attributed to contamination from the equipment used for UV irradiation. The XPS technique typically probes the upper ~20–30 nm of the thin film samples. The concentration differences could be further attributed to the diffusion movement and redistribution during annealing as well as to the different level of the surface contaminations induced after postdeposition treatments. Moreover, the oxygen vacancies’ concentration changes with these treatments.

### 3.3. AFM Analysis

The surface morphologies of the thin film samples were examined by means of an AFM analysis with a scanning area of 20 × 20 μm^2^. The 2D AFM images of the samples are shown in Figure 3. All of the films possess good uniformity over the complete scanning area. The root-mean square (RMS) roughness of each sample is summarized above the 2D AFM images. It was found that AZO after UV–ozone treatment possesses a smoother surface. The smoothness of a TCO film is an indispensable property for flat panel displays and piezoelectric devices. The root mean square (RMS) roughness should be less than a particular value, usually 2 nm for these applications. The RMS roughness of the annealed AZO presents a slightly increasing trend compared to the as-deposited one, reflecting the increased average grain size which was also observed through XRD analysis.

### 3.4. Optical Properties

The optical transmittance spectra of the AZO films as deposited on glass, after UV–ozone, and after thermal annealing treatments are shown in Figure 4a. The spectrum of commercially available ITO on a glass substrate is also included for reference. As seen, the transmittance of the as-deposited AZO film and AZO film after postdeposition UV–ozone treatment are almost identical; however, a slight transmittance decrease was measured for the annealed AZO films. The absorption edge of the UV–ozone-treated sample is almost the same as the deposited AZO film, while, for the annealed AZO film, it has shifted to the longer wavelengths. This “red-shift” corresponds to a decrease in carrier concentration during the annealing process in the air atmosphere. The result illustrates that the oxygen vacancies decrease in the AZO films after annealing in an air atmosphere, which supports the XPS data.

Optical band gap (E_g_) was calculated from the absorption coefficient (α) counting the thickness of the film (d) and transmittance percentage (T) using:(1)$$\alpha =2.303\frac{\log\left(\frac{1}{T}\right)}{d}$$
and the Tauc relation [37].

Figure 4b shows a plot of (αhυ)^2^ versus photon energy (hυ), where an extrapolation of the linear region indicates an approximation of the optical band gap. The calculated E_g_ value for AZO thin films varies depending on the post-treatment conditions from ~3.11 eV (for thermally annealed sample) to ~3.14 eV (for UV–ozone). The measured band gap value of the as-deposited AZO thin film of 3.15 eV is in good agreement with that reported in the literatures [8,38,39,40,41]. For AZO films, band gap variation is usually associated with the carrier concentration. The carrier density in AZO thin films depends on the generation of zinc interstitial atoms and the formation of oxygen vacancies. In our study, all AZO film samples were fabricated in one batch; therefore, there is no possibility of changes in Zn content and the overall films’ composition stoichiometry (as discussed before the surface composition could be modified by postdeposition treatments). The postdeposition treatment processes were completed using identical samples. Therefore, any changes in the carrier concentration of these AZO films can be attributed solely to the formation (increase) or extinction (decrease) of oxygen vacancies. As electrical conductivity is directly dependent on the carrier concentration of the material, the total amount of oxygen vacancies plays a significant role in overall thin film performance. The optical bandgap energy of post-treated samples could further decrease due to a decline in the Al concentration in the lattice, which is attributed to the formation of the metastable aluminum-oxide phase during annealing [42]; however, the XRD data ruled out this scenario.

### 3.5. Wettability Properties

Generally, the quantitative measure of wettability is the contact angle θ, geometrically determined by a liquid at the three phases’ boundary intersection between the liquid, gas, and solid [43]. Upon determining the contact angle, the surface energy (a key parameter characterizing the solid–liquid interface that is closely related to the adhesion) can be calculated using the Owen–Wendt–Rabel and Kaelble (OWRK) method [44]. To apply the OWRK method, two types of liquids with known dominant dispersion and polar components are usually necessary. In our case, we used deionized (DI) water and glycerol, respectively.

Figure 5 shows the optical images of a water and glycerol droplet on an AZO surface at the three states: after UV–ozone, after annealing treatments, and as deposited. The contact angle of the UV–ozone treated and annealed AZO films measured for DI water decrease while in contact with glycerol and increase compared with the as-deposited AZO film. The high contact angle of the AZO surface with DI water indicates its hydrophobicity, which is not affected by the surface treatment [35]. The measured high contact angle values confirm its poor adhesion/wettability. This is in accordance with the ZnO, which shows a hydrophilic surface in nature due to the weak adhesive van der Waals force between ZnO and water droplet [45,46].

All measured parameters related to the surface energy calculations are shown in Table 4. The surface energy values decrease after postdeposition treatment, most significantly in the case of the UV–ozone-treated AZO film, due to the smoother surface (high concentration of oxygen vacancies). In case of contact with DI water, the surface tension value is 72.8 mN/m with polar and dispersed values of 51mN/m and 21.8 mN/m, respectively. In case of glycerol, the surface tension value is 63.4 mN/m with polar and dispersive values of 26.4 mN/m and 37 mN/m, respectively.

### 3.6. Electrical Properties

The sheet resistance, resistivity, conductivity, carriers’ concentration, and mobility, determined by Hall-effect measurements, are shown in Table 5. The electrical characterization reveals that the conductivity of the AZO films increases after UV–ozone postdeposition treatment and that the sheet resistance decreased from 66 Ω.cm^−1^ (in the case of the as-deposited AZO) to 56 Ω.cm^−1^ (after UV–ozone treatment), which is attributed to an increase in the oxygen vacancies’ concentration. At the same time, there are no changes in the structure and the optical transmittance in comparison with as-deposited AZO films. A similar trend has been reported in [47]. Lowering the sheet resistance without causing an adverse effect on the optical properties is important, and this is achieved by applying a noninvasive and mild treatment such as the UV–ozone procedure.

The thermal annealing also decreases the sheet resistance values compared to the as–deposited AZO. It has been observed that the thermal annealing tends to remove the oxygen vacancies and improve the crystalline structure as also reported in [44].

Larger grain sizes and an improvement in the crystal structure of AZO with annealing were observed. In addition to the crystalline structure boost, the annealing treatment facilitates the oxygen atoms/molecules’ diffusion from the air into the AZO films in order to fill the oxygen vacancies. A better crystallinity of the films should result in a scattering center for electrons reduction, and, consequently, an increase in carrier mobility. It is well known that the oxygen vacancies induce stress to the lattice structure [34]. Hence, the oxygen vacancies’ defects reduction through high-temperature annealing releases the stress [48]. The UV–ozone treatment shows no significant modification of grain size and crystallinity. Larger grain sizes lead to fewer grain boundaries, which act as traps and obstacles for free carriers. As a result, the large-grain AZO films obtained after annealing possess both better conductivity and improved optical properties.

Recently, it has been reported that the elimination of the oxygen vacancies in AZO thin films reduces the carrier concentration and consequently shifts the Fermi level toward the valence band, resulting in a decrease in the bandgap [24].

It was found that the UV–ozone treatment of AZO films improves the conductivity and lowers the sheet resistance. The conductivity of the AZO films depends on the oxygen vacancy concentration in addition to the aluminum doping. Oxygen vacancies give rise to the charge carriers and dangling Zn bonds, thus improving the conductivity [32]. It was further reported that the oxygen deficiency, i.e., oxygen vacancies, enhances the carrier concentration by a factor of two since the oxygen vacancy can provide carriers with donation sites [49].

## 4. Conclusions

AZO films with preferable (100) orientation are deposited using ALD. The postdeposition treatment with UV–ozone exposure and annealing at 600 °C leads to modifications in the films’ structure, chemical states (oxygen vacancies), and surface morphology. Both treatments lead to improved electrical characteristics. After the UV–ozone treatment, the changes in the crystal structure, surface morphology, and optical property of the thin films were not significant; however, the concentration of the oxygen vacancies in the thin films increased. As a result, a decrease in the sheet resistance and an improvement in the conductivity were attributed to an increase in the oxygen vacancy concentration, while the improved conductivity of the annealed AZO films was due to the crystalline structure/grain size enhancement. The variation of the O_v_/O_tot_ ratio indicates the high tunability of the film properties depending on the post deposition treatment. This is an important achievement realized by using no invasive treatment. The postdeposition treatment modifications are beneficial for the opto-electrical properties and for wettability optimization.

## Figures and Tables

**Figure 1 nanomaterials-13-00800-f001:**
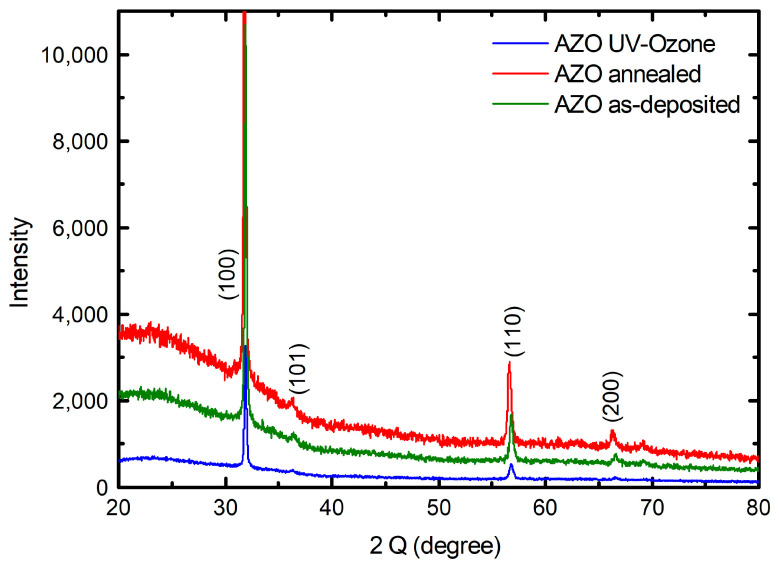
XRD patterns of AZO layers at three different states: UV–ozone treated, annealed state, and as-deposited.

**Figure 2 nanomaterials-13-00800-f002:**
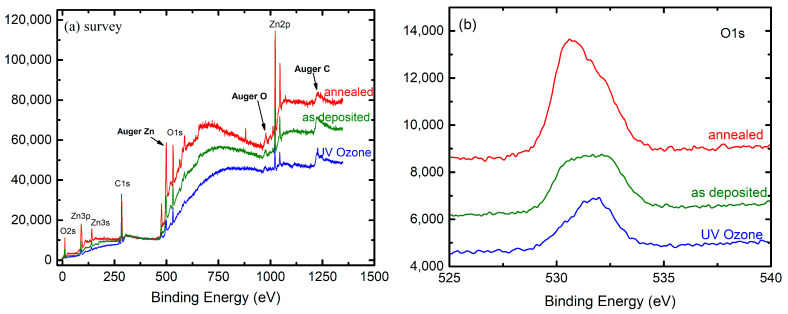
(**a**) The survey XPS spectrum of AZO film as deposited, after UV–ozone, and after annealing treatment; (**b**,**c**) XPS spectra of O1s and Zn2p; (**d**–**f**) deconvolution O1s peaks of AZO (UV–ozone), AZO (annealed), and AZO (as-deposited) layers.

**Figure 3 nanomaterials-13-00800-f003:**
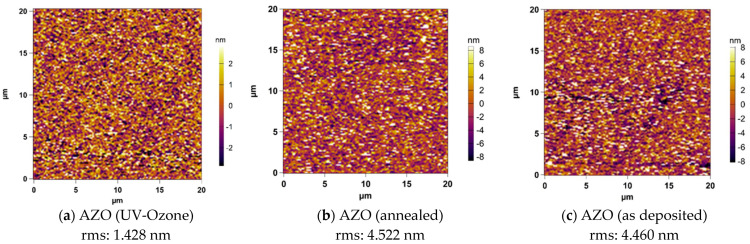
AFM 2D images and surface roughness values of AZO/glass substrates with postdeposition treatment: (**a**) UV–ozone, (**b**) annealed and (**c**) as-deposited AZO films.

**Figure 4 nanomaterials-13-00800-f004:**
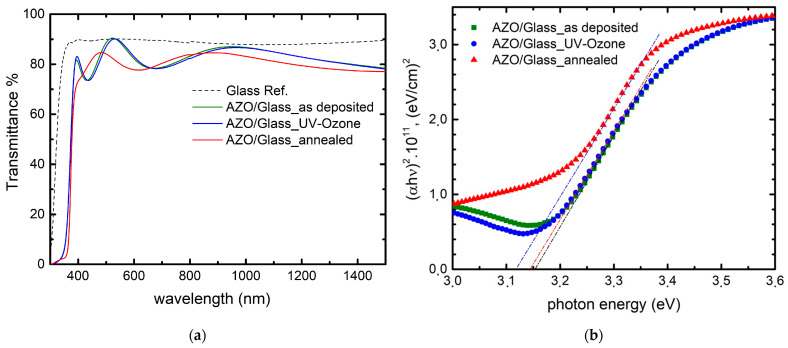
(**a**) Transmittance spectra of AZO (UV–ozone), AZO (annealed), and AZO (as deposited) and (**b**) plot of (αhυ)^2^ versus photon energy (hυ) of AZO (UV–ozone) and AZO (annealed).

**Figure 5 nanomaterials-13-00800-f005:**
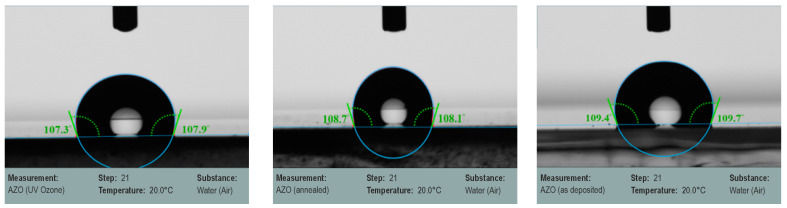
Contact angle of AZO films on glass substrates measured for deionized (DI) water (**top row**) and glycerol (**bottom row**) droplets. Left to right: AZO (UV–ozone), AZO (annealed), and AZO (as deposited).

**Table 1 nanomaterials-13-00800-t001:** Unit cell parameters and mean crystallite size.

Sample	a [Å]	c [Å]	Mean Size [nm]
AZO (UV–ozone)	3.244(4)	5.212(3)	66 ± 2
AZO (annealed)	3.252(3)	5.197(4)	73 ± 2
AZO (as deposited)	3.245(4)	5.219(3)	71 ± 2

**Table 2 nanomaterials-13-00800-t002:** The variation of the O_v_/O_tot_ ratio.

	AZO (UV–Ozone)	AZO (Annealed)	AZO (as Deposited)
**O_v_/O_tot_**	0.39	0.29	0.32

**Table 3 nanomaterials-13-00800-t003:** Calculated surface concentrations.

Sample	C, at.%	O, at.%	Al, at.%	Zn, at.%
**AZO (UV–ozone)**	83.67	11.55	0.67	4.11
**AZO (annealed)**	53.63	28.41	1.44	16.52
**AZO (as deposited)**	72.50	18.83	0.93	7.74

**Table 4 nanomaterials-13-00800-t004:** Contact angle, SFE and SFT of UV–ozone treated, annealed, and as-deposited AZO films.

			SFE [mN/m]		
Sample	Liquid	CA [°]	Total SFE [mN/m]	Polar [mN/m]	Dispers [mN/m]
**AZO (UV–ozone)**	water	108.12 ± 0.34	9.06 ± 0.95	4.63 ± 0.54	4.43 ± 0.41
	glycerol	104.24 ± 0.58			
**AZO (annealed)**	water	109.37 ± 0.45	10.85 ± 0.93	8.64 ± 0.63	2.2 ± 0.30
	glycerol	101.26 ± 0.40			
**AZO (as deposited)**	water	110.69 ± 0.90	14.69 ± 1.54	13.96 ± 1.24	0.73 ± 0.30
	glycerol	98.32 ± 0.43			

**Table 5 nanomaterials-13-00800-t005:** Electrical properties of UV–ozone, annealed, and as-deposited AZO films.

Electrical Properties	AZO (As Deposited)	AZO (UV–Ozone)	AZO (Annealed)
**Sheet resistance (Ω.cm^−1^)**	66.00	56.12	59.52
**Resistivity (Ω.cm)**	1.78 × 10^−3^	1.52 × 10^−3^	1.61 × 10^−3^
**Conductivity (1/Ω.cm)**	561.37	660.00	622.00
**Mobility (cm^2^/V.s)**	−9.27	−7.94	−8.81
**CCC Sheet (1/m^2^)**	−1.02 × 10^20^	−1.40 × 10^20^	−1.19 × 10^16^
**CCC Bulk (1/cm^3^)**	−3.78 × 10^20^	−5.19 × 10^20^	−4.41 × 10^20^
**Avg. Hall coefficient (cm^3^/C)**	−16.51 × 10^−3^	−12.03 × 10^−3^	−14.15 × 10^−3^

## Data Availability

Not applicable.

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
