# Peer review of "The Effect of Post Deposition Treatment on Properties of ALD Al-Doped ZnO Films"

_nanomaterials, 2023, doi:10.3390/nano13050800_

Round 1
Reviewer 1 Report
This work reported the treatment of Al-doped ZnO thin films after ALD deposition and compared the impact of different treatment methods (UN-ozone and thermal annealing) on the film properties. The authors showed the effective tuning on the film structure, electrical and optical properties. According to my opinion, this work should be published in Nanomaterials after the following issues are address:
1. Page 3, Figure 1
a. The films showed big differences in XRD peak intensity. Could the authors add some discussion on it?
b. The minor peak at ~ 67 deg should be marked and discussed
2. Page 4, Table 1
a. The change of the lattice parameters (drift of peak position in XRD) might indicate the formation of the stress in the film. Would this be related to the film property?
3. Page 5, Table 3
a. The Al and Zn also showed big concentration change after treatment. The discussion/explanation on such result is missing
4. Page 6, line 215
a. The authors claimed the change of the optical band gap is due to the decrease of the Al concentration in the film. While the more dominant component Zn showed even more composition change in Table 3. Would the Zn composition change potentially impact the optical band gap?

Author Response
Reviewer 1
We would like to thank the Reviewer #1 for very useful and important comments aiming at the improvement of our manuscript. We take into consideration all the comments, questions and suggestions during the revision process, marked in red in the revised manuscript.
Comments and Suggestions for Authors
This work reported the treatment of Al-doped ZnO thin films after ALD deposition and compared the impact of different treatment methods (UN-ozone and thermal annealing) on the film properties. The authors showed the effective tuning on the film structure, electrical and optical properties. According to my opinion, this work should be published in Nanomaterials after the following issues are address:
- Page 3, Figure 1
- The films showed big differences in XRD peak intensity. Could the authors add some discussion on it?
All the samples do not show any peaks related to Al-related oxides, and this suggests that Al ions are incorporated into the wurtzite type structure. The narrowing of the dominant (100) peak as well as intensity increase after annealing indicates the improved crystallinity.
- The minor peak at ~ 67 deg should be marked and discussed
addressed in the text and figure
- Page 4, Table 1
- The change of the lattice parameters (drift of peak position in XRD) might indicate the formation of the stress in the film. Would this be related to the film property?
The (100) peak 2Q value shifts from 31.89° to 31.79° after post-deposition annealing, while changing only slightly to 31.87 after UV-ozone treatment. In addition to the repair of the crystalline structure, the annealing treatment allows oxygen atoms/ molecules to diffuse from the air into the Al: ZnO films to fill the oxygen vacancies. As oxygen vacancies are known to induce stress to the lattice structure [J. Alloy. Compd. 2016, 688, 122–132], the reduction of the oxygen vacancies through annealing releases the stress. The opposite happens after UV-ozone post- deposition treatment: according to the XPS data oxygen vacancies increases. The calculated microstrain of as deposited films was 7.7x10-4, which are decreasing to 7.5x10-4 after annealing and increased to 8.3 x 10-4 after UV-ozone treatment respectively as follows of the oxygen vacancies and mean crystalline size trends.
The number of carriers increases by the UV-ozone treatment. This phenomenon may be related to a photocatalytic effect on the film surface. The mechanism of the photocatalytic effect of ZnO:Al is as follows: UV photons are absorbed by ZnO through the band-gap transition, generating electron-hole pairs.
Then, the generated electrons or holes react with O2 or H2O in air to form active oxygen species, additionally O3 is released in UV radiation. Furthermore, the residual organics in the films are decomposed by the active oxygen species, leaving oxygen vacancies in the ZnO lattice through the thermal desorption of carbon or hydrogen species. The cleavage of ZnO bonds (photolysis) by the charge transfer from O2- to Zn2+ also contributes to the formation of oxygen vacancies. The number of electrons in the conduction band increase by increasing the number of oxygen vacancies.
- Page 5, Table 3
- The Al and Zn also showed big concentration change after treatment. The discussion/explanation on such result is missing
XPS technique typically probes the upper ~20–30 nm of the thin film samples. The concentrations difference could be attributed to the diffusion movement and redistribution during annealing as well as different level of the surface contaminations induced after post-deposition treatments. Moreover, the oxygen vacancies concentration changes with these treatments.
- Page 6, line 215
- The authors claimed the change of the optical band gap is due to the decrease of the Al concentration in the film. While the more dominant component Zn showed even more composition change in Table 3. Would the Zn composition change potentially impact the optical band gap?
For ZnO:Al (AZO) films, band gap variation is usually associated with the carrier concentration. Generally, there are two primary mechanisms which control carrier density in AZO thin films: (i) the generation of zinc interstitial atoms and (ii) the formation of oxygen vacancies [Applied Surface Science, Volume 484, 2019, pp. 990-998]. In our study all AZO film samples were fabricated in one batch therefore there is no possibility of changes in the Zn content and overall films composition stoichiometry (as discussed before the surface composition could be modified by post-deposition treatment). The post deposition treatment processes were completed using identical samples. Therefore, any changes in carrier concentration of these AZO films can be attributed solely to the formation (increase) or extinction (decrease) of oxygen vacancies. As electrical conductivity is directly dependent on the carrier concentration of the material, the total amount of oxygen vacancies plays a significant role in overall thin film performance.
The air-annealing promotes oxygen in the air to fill native oxygen vacancies within the film, resulting in an overall decrease in carrier concentration and a Fermi shift towards the valence band. This was confirmed through XPS core-level high resolution analysis, where a distinct shift from an oxygen deficient- to an oxygen sufficient-state was observed in the O 1s peaks with annealing. Oxygen vacancies however increase after UV-ozone treatment as stated above. For further verification of this trend, UV–vis spectroscopic analysis of the AZO films exhibited a decrease in the optical band from 3.15 eV of as-deposited AZO to 3.11 eV with annealing, suggesting a decrease in carrier density due to the Burstein-Moss effect.

Reviewer 2 Report
This manuscript by Petrova et al reports on the properties of Alumimum-doped zinc oxide grown by atomic layer deposition after post-deposition treatment. In particular, two different treatments were applied: UV ozone and thermal annealing. Topography (roughness) was inspected by AFM, while XPS was used to determine chemical composition and XRD to study the crystallinity. Additionally, wettability and optical properties were analysed.
The manuscript presents a complete work and it is in general well organized and presented although writing should be checked in order to correct some mistakes and typos. Additionally:
1. Although it presents original results, reference to works previously published reporting similar studies should be done and discussion is needed. For instance Kuprenaite et al, Thin Solid Films 599, 19-26 (2016), Geng et al, ECS Journal of Solid Science and Technology 1 N45 (2012), Singh et al, Journal of Nanoscience and Nanotechnology 16, 861 (2016).
2. The wettability properties are not properly presented:
- On the one hand the results obtained should be related to the results obtained by other techniques, since contact angle values are related to the roughness and/or the surface chemistry.
- The reason for choosing the liquids is not clear. In fact, is glycerol relay apolar?
- References about the values of surface tension of the liquids and the corresponding components in needed.
- Details on how the surface energy is calculated (what model is used) are needed.
Author Response
Reviewer 2
We would like to thank the Reviewer #2 for very useful and important comments aiming at the improvement of our manuscript. We take into consideration all the comments, questions and suggestions during the revision process, marked in red in the revised manuscript.
Comments and Suggestions for Authors
This manuscript by Petrova et al reports on the properties of Alumimum-doped zinc oxide grown by atomic layer deposition after post-deposition treatment. In particular, two different treatments were applied: UV ozone and thermal annealing. Topography (roughness) was inspected by AFM, while XPS was used to determine chemical composition and XRD to study the crystallinity. Additionally, wettability and optical properties were analysed.
The manuscript presents a complete work and it is in general well organized and presented although writing should be checked in order to correct some mistakes and typos. Additionally:
- Although it presents original results, reference to works previously published reporting similar studies should be done and discussion is needed. For instance Kuprenaite et al, Thin Solid Films 599, 19-26 (2016), Geng et al, ECS Journal of Solid Science and Technology 1 N45 (2012), Singh et al, Journal of Nanoscience and Nanotechnology 16, 861 (2016).
These references are included in the references list and discussion inserted where relevant in the manuscript text
- The wettability properties are not properly presented:
-On the one hand the results obtained should be related to the results obtained by other techniques, since contact angle values are related to the roughness and/or the surface chemistry.
The wettability of material surfaces depends on the surface roughness (morphology) and the chemical composition altering the surface free energy. Roughness affects the wettability of a solid and thus the contact angle of a liquid as well as adhesion. A rule of thumb states that roughness improves wettability at small contact angles (angle becomes smaller), and reduces it at large contact angles (angle becomes larger). The limiting contact angle of wettability, which determines the deflection in one direction or the other, is 90°. The wettability of studied AZO films depends on the oxygen vacancies density. ZnO surfaces become more hydrophilic with increasing oxygen vacancies density.
-The reason for choosing the liquids is not clear. In fact, is glycerol relay apolar?
In the OWRK method, two liquids with known dispersion and polar components are required to determine the surface energy. One liquid must have a dominant polar component and the other a dominant dispersive component. The liquids we use are deionized water and glycerol. The values of the polar and dispersed part of the surface tension of deionized water are 51 mN/m and 21.8 mN/m, respectively. For glycerol, the difference between the values of polar and dispersed part of surface tension is not so clearly expressed. These are 26.4 mN/m and 37 mN/m, respectively. However, the dispersion component is dominant. Therefore, the glycerol is a relatively polar solvent, but less polar than water.
-References about the values of surface tension of the liquids and the corresponding components is needed.
The surface tension values of DIW and glycerol are 72,8 mN/m and 63,4 mN/m respectively. These are included in the manuscript.
-Details on how the surface energy is calculated (what model is used) are needed.
The Owen–Wendt–Rabel - Kaelble (OWRK) method was used to determine SFE. It divides the interactions at the liquid-solid interface into dispersive and polar. The surface energy of the deposited AZO layers were measured with a Drop Shape Analyzer measuring system (DSA25S, KRÜSS GmbH, Hamburg, Germany). The method idea is that the magnitude of the liquid-solid surface tension is determined by various interfacial interactions that depend on the properties of both the liquid being measured and the solid surface. The polarity of DIW is 1.0 while those of glycerol 0.812. The viscosity of both liquids should be also mentioned in context to liquids choice for the measurements. At 20 C the viscosities are as follow: DIW 10-3 Pa.s , and glycerol 1.412 Pa.s respectively.
Round 2
Reviewer 2 Report
There are no additional comments after the revision of the manuscript.